# Effects of Gas-Volume Fractions on the External Characteristics and Pressure Fluctuation of a Multistage Mixed-Transport Pump

**Chenhao Li [1], Xingqi Luo [1], Jianjun Feng [1,\*], Guojun Zhu [1] and Yangang Xue [2]**

[1] State Key Laboratory Base of Eco-Hydraulic Engineering in Arid Area, Institute of Water Resources and Hydropower, Xi'an University of Technology, Xi'an 710048, China; drleech_xaut@126.com (C.L.); luoxq@xaut.edu.cn (X.L.); guojun_zhu1984@126.com (G.Z.)

[2] LanZhou Institute of Technology, No.1 Gongjiaping East Road Qilihe District, Lanzhou 730050, China; xyg3024_cn@163.com

\* Correspondence: jianjunfeng@xaut.edu.cn; Tel.: +86-188-0293-6315

**Abstract:** In the petroleum industry, multiphase transport pumps are a crucial technology for petroleum extraction. Therefore, the pressure fluctuation and internal-flow characteristics of multiphase transport pumps with different inlet-gas-volume fractions (IGVFs) have become an important research topic. Studying the pressure fluctuation and its effects on the performance of mixed-transport pumps under different IGVFs is significant for improving the running stability of pumps. In this work, steady and transient flow with different IGVFs was solved using the Navier–Stokes equation applied to a structured grid and the Shear Stress Transport (SST) turbulence model. The effects of IGVFs on the pressure pulsation and the performance of a three-stage, mixed-transport pump were studied. Results indicated that the numerical calculation results agreed well with the experimental data. The numerical method could predict the gas and liquid two-phase flow in the mixed-transport pump accurately. The pressure increase of this pump decreased with the increased flow quantity and the IGVFs. The efficiency improvement of the pump was limited by the increasing the flow rate. Under the rated-flow condition, a quantitative relationship was established between the relative discharge of the IGVF and the decrease in the pump head; when the IGVF exceeded 15%, the pressurization capacity decreased by more than 30%. Along the blade centerline direction, the pressure fluctuation amplitude near the suction surface of the impeller blade head gradually increased. Numerical simulation results showed that the dominant frequency of the pressure fluctuation of the impeller and diffuser was ten and seven times that of the rotation frequency, respectively. Thus, the IGVFs greatly influenced the dominant frequency of the pressure fluctuation. The air in the impeller primarily piled up at the suction surface of the blade head near the front cover. Under a centrifugal force, water was pushed to the back cover plate, making the gas-volume fraction near the front cover plate higher. Consequently, the distribution of gas content in the impeller became uneven. On the blade suction surface near the front cover plate, a low-velocity area caused by flow separation was generated, which further affected the pressure pulsation in the impeller. There were obvious vortices in the diffuser, and the vortex position had a tendency to move toward the inlet of the diffuser with an increased gas content. The flow pattern in the impeller was consistent, which indicated the great transport performance of this pump. In conclusion, through numerical simulation and experimental research, this study revealed the effects of the IGVFs on the performance and pressure pulsation of a mixed-transport pump under a gas–liquid two-phase flow condition. Our findings may serve as a guide for the optimization of a multiphase pump.

**Keywords:** deep-sea multiphase pump; energy characteristics; gas–liquid two-phase flow; pressure fluctuation; gas-volume fraction

## 1. Introduction

The economic and industrial high-speed development has led to remarkable changes to people's daily lives and the environment. Therefore, reducing the environmental pollution caused by resource exploitation and improving production efficiency has become a momentous subject. Marine oil resources are abundant and exploration is urgently needed. The multiphase pump has become an important piece of production equipment for the efficient transportation of non-renewable resources in the energy industry. The application of oil–gas mixed-transportation technology can considerably simplify the marine treatment process, reduce the platform area, and reduce the cost of construction investment by 40% [1].

The flow inside a multiphase mixed-transport pump is complicated, and the performance of the pump is affected by many factors, such as loss of friction, erosive environment, divulgation loss, and different inlet gas contents [2,3]. The prediction of the performance and dependability can be validated by analyzing the lift and efficiency of a mixed-transport pump under different inlet-gas-volume fraction (IGVF) conditions [4]. Li et al. [5] adopted a new type of underwater gas–liquid mixer to reduce the fluctuation range under the gas content condition and improve the boosting capacity of the mixed transfer pump. Shi et al. [6] studied the turbulence intensity and turbulence dissipation law within the compression stage of the mixed-transport pump under the air and water two-phase flow condition. Yu et al. [7] showed that the forces between the air and water in the gas and liquid flow include a lift force, a turbulent dissipation force, resistance, and a virtual mass force. Under transient conditions, the pump head decreases and instantaneous fluctuation occurs, given the effect of a virtual mass force. Therefore, studying the pressure pulsation and internal flow field of the multistage mixed-transport pump under different gas content conditions is necessary.

Pressure pulsation exists in various types of pumps as a universal phenomenon. Studies on the mixed-transport pump were conducted through experiments and numerical calculations. Rabiger et al. [8] used pressure sensors to measure transient pressure signals under 90% and 98% IGVF condition for analysis, as well as pressure fluctuation. Yu et al. [9] found that the pressure fluctuation in the inlet area was intense by exploring the transfer process of the mixed-transport pump. Yu et al. [9] also discovered that under different IGVFs, the head of the multistage mixed-transport pump decreased first and then fluctuated periodically. Serena et al. [10] studied the transient phenomenon of a mixed-transport pump through experiments; their findings showed that the pressure pulsation amplitude was higher and the frequency was lower when the flow rate was small. Zhang et al. [11] studied gas aggregation, gas–liquid separation, and pressure fluctuation at different positions in the mixed-transport pump under a gas–liquid two-phase flow. They showed that axial power, pressure, and liquid phase volume fractions fluctuated during the experiment. Tan et al. [12] studied the T-shaped blade pressure pulsation under mixed flow and discovered that the pressure fluctuation amplitude was the largest at the blade head. Moreover, the effects of the blade setting angle and tip clearance on pressure fluctuation and flow law were studied.

In recent years, many investigations have been carried out on the internal flow mechanism and flow pattern of the mixed-transport pump. Zhang et al. [13] revealed the internal stream field specifics of a vane-type gas and liquid mixed-transport pump. Zhang et al. [14] improved the gas-volume fraction (GVF) distribution at the hub by optimizing the shape of the multiphase pump; their findings suggest that with an increased gas–liquid separation, the liquid-phase medium in the impeller flows near the blade pressure surface and the gas phase gathers at the low-pressure surface side of the blade. Zhang et al. [15] found that the little bubbles are round while the large bubbles are ellipsoidal by studying the movement law of bubbles. The size and number of bubbles increase with an increased IGVFs. Zhang et al. [16] established that the gas volume distribution gradually decreases from the inlet suction to the outlet suction. Given the centrifugal force influence, the GVF at the 90% blade height of the impeller is low. Kim et al. [17] found through numerical analysis that the bubbles in the multiphase pump break continuously with an increasing pressure; under the action of centrifugal force, the liquid was pushed to the shroud of the diffuser and flow separation was suppressed by

optimizing the multiphase transport pump. The visualization experiment of Serena et al. [18] on the instability mechanism of the multiphase transport pump suggests that air accumulated on the blade suction surface side under a low-pressure condition.

Studying the pressure fluctuation of a multistage mixed-transport pump and its effects on the performance of a multiphase pump under different IGVFs is important. As such, the operational stability and efficiency of the multiphase pump can be improved. In this research, the pressure fluctuation and the performance change of a multistage mixed-transport pump under different IGVFs were studied via numerical simulation and experiments. The findings may serve as a guide for the optimal design of a multiphase pump.

## 2. Model and Mesh Generation

### 2.1. Physical Model

A new multistage mixed-transport pump was designed based on the research project of offshore oil and gas production and transportation multiphase mixed-transport pump of the National Natural Science Foundation of China (NSFC). The geometric parameters and design point data of this pump are shown in Table 1. Furthermore, Figure 1 shows the Computational Fluid Dynamics (CFD) numerical simulation calculation domain between the inlet and the outlet of the three-stage, deep-sea mixed-transportation pump. The pump is composed of an inlet pipe, an impeller, a cavity connection section, a diffuser, and an outlet pipe. The impeller and diffuser of the three-stage, deep-sea mixed-transportation pump are identical. The numerical simulation of the three-stage mixed-transportation pump was conducted under a pure water condition and different IGVFs conditions.

**Table 1.** Design parameters and geometric dimensions of a single-stage multiphase pump.

| Component | Designation | Parameters |
|---|---|---|
| Design point | Rated discharge ($Qd$, m$^3$/h) | 26.5 |
| | Rated lift ($Pr$, KPa) | 260 |
| | Rated revolution ($n$, r/min) | 3500 |
| Impeller | Number of blade ($Zi$) | 7 |
| | Inlet inner diameter ($D1$, mm) | 62.23 |
| | Outlet inner diameter ($D2$, mm) | 127 |
| Diffuser | Number of vane ($Zd$) | 10 |
| | Inlet inner diameter ($D3$, mm) | 150 |
| | Outlet inner diameter ($D4$, mm) | 65.49 |

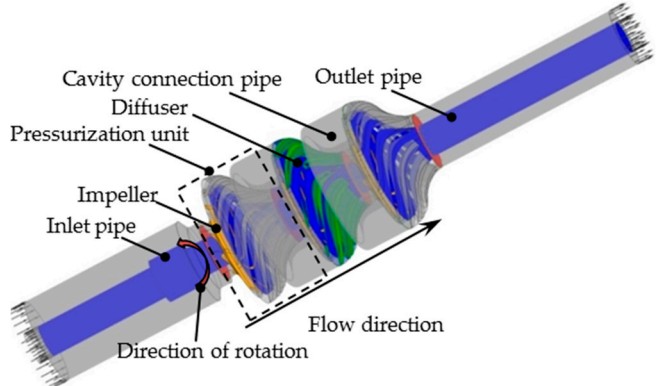

**Figure 1.** Calculation domain of the three stages of the multiphase pump.

## 2.2. Grid Division and Monitoring Point Setting

ANSYS ICEM (ANSYS 18.0 PA USA) as used to draw the grid for the inlet and outlet pipe and cavity connection structure to ensure the accuracy and good convergence of the numerical simulation. Turbo-Grid (ANSYS 18.0 PA USA) was used for the grid division of the impeller and the diffuser. A structured hexahedral grid was used for each component. Then, mesh densification was carried out on the parts near the wall surface and the excessive curvature in the main flow components. The grid of a single channel was established for the impeller and the diffuser. The grid was generated via the periodic rotation around the Z-axis. The structured grid of the pump impeller and diffuser is shown in Figure 2, among which, the O-block topological structure was used for refinement to achieve an efficient simulation.

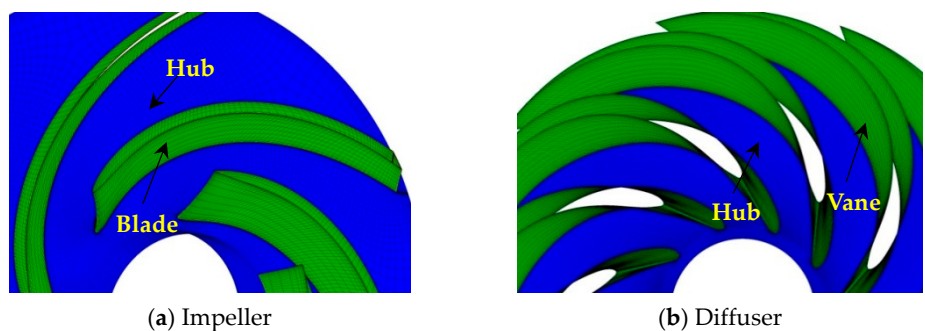

(**a**) Impeller　　　　　　　　　　　　　　　　(**b**) Diffuser

**Figure 2.** Grids of the impeller and the diffuser.

Figure 3 shows the setting positions of the pressure data acquisition point in an unsteady numerical simulation of the multiphase mixed-transportation pump. Five pressure monitoring points (IMS1–IMS5) were set on the pressure side of the impeller blade. Five pressure monitoring points (DPS1–DPS5) were set on the pressure side of the diffuser guide vane.

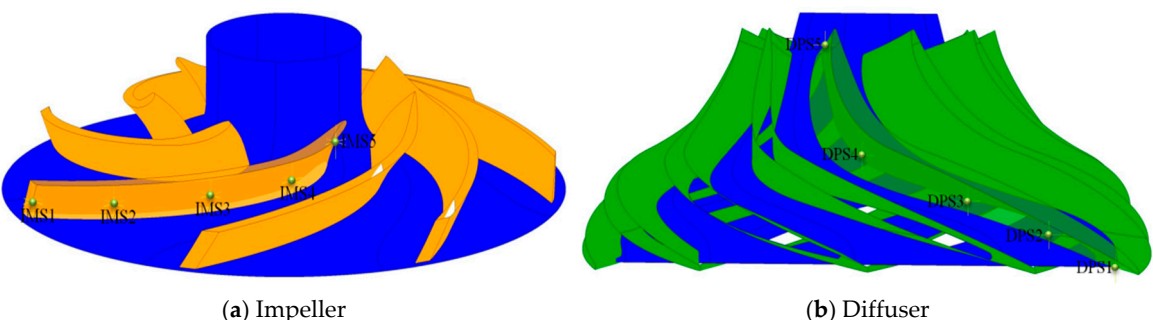

(**a**) Impeller　　　　　　　　　　　　　　　　(**b**) Diffuser

**Figure 3.** Location of the monitoring point.

## 3. Numerical Methodology and Experimental Verification

### 3.1. Numerical Methods and Solution Settings

Computational fluid dynamics software ANSYS-CFX18.0 was used to simulate the three-stage mixed-transport pump. The water (liquid phase) and air (gas phase) were respectively treated as a consecutive medium fluid and a dispersed fluid. In this calculation, considering the transfer of turbulent shear stress, the SST turbulence model, which can accurately predict the flow separation under a reverse pressure gradient, was used for the continuous fluid (water), while the dispersed fluid (air) was taken in the dispersed phase zero equation model.

The boundary conditions for the calculations were set as follows: the inlet was set to have a static pressure and the outlet was given a mass flow rate in accordance with the experiment data, and the physical walls of the mixed-transport pump were equipped with a non-sliding wall. The

frozen rotor methods and transient rotor stator method were respectively adopted for the steady-state and transient numerical calculations to couple the stationary and the rotating parts. The convergence condition of the steady-state numerical calculation was set to an RMS value less than $10^{-4}$. Then, the steady-state simulation outcomes were obtained as the original value for the transient numerical calculation [19,20]. In accordance with the impeller rotation speed, the time step was determined to be $Ti = 60/(3500 \times 180) = 9.5238 \times 10^{-5}$ s. The different IGVFs were set at the inlet during the numerical simulations. For example, when the IGVFs = 5%, the inlet gas content was set to 0.05 and the liquid volume fraction was set to 0.95.

### 3.2. Mesh Validation

A reasonable number of grids for numerical simulation should be selected to save time and computing resources. Therefore, five groups of grids were used in the grid independence test. The total number of grids in each computing domain ranged from 3,072,474 to 9,283,628. Table 2 shows the number of cells in each computing domain. The boost equation is defined as:

$$P_b = P_{out} - P_{in}, \tag{1}$$

where $P_{out}$ represents the outlet pressure of the mixed-transport pump and $P_{in}$ represents the inlet pressure.

**Table 2.** Supercharging and efficiency in relation to mesh elements.

| Items | Mesh I | Mesh II | Mesh III | Mesh IV | Mesh V |
|---|---|---|---|---|---|
| Total grid number | 3,072,474 | 5,378,384 | 6,604,412 | 7,734,156 | 9,283,628 |
| Pressure rise (kPa) | 776.12 | 784.52 | 788.61 | 789.20 | 789.10 |
| Efficiency (%) | 78.25 | 80.45 | 80.51 | 80.57 | 80.59 |
| Relative pressure rise *Pr/Pd1* | 1 | 1.0108 | 1.0161 | 1.0168 | 1.0167 |
| Relative efficiency $\eta/\eta c$ | 1 | 1.0025 | 1.0032 | 1.0040 | 1.0041 |

Efficiency $\eta$ is defined as the ratio of the actual pump head $H_a$ and the theoretical calculation head $H_t$. The expression is given as follows:

$$\eta = H_a/H_t, \tag{2}$$

where the actual pump head $H_a$ is defined as:

$$H_a = (P_{out} - P_{in})/\rho_{mix}g, \tag{3}$$

$$\rho_{mix} = IGVF \times \rho_{gas} + (1 - IGVF) \times \rho_{water}, \tag{4}$$

where $\rho_{mix}$, $\rho_{gas}$, and $\rho_{water}$ are the gas–liquid mixed density, the gas-phase density, and the liquid-phase density, respectively. g represents acceleration due to gravity.

Moreover, the theoretical head $H_t$ is defined as follows:

$$H_t = (M \times \omega)/\rho_{mix} \times Q \times g, \tag{5}$$

where $M$ represents the torque, $\omega$ represents the angular velocity, and $Q$ represents the volume flow rate.

It can be obtained from the calculation results shown in Figure 4 for the mesh listed in Table 2 that after grid IV, the diversity in the pressure rise and the efficiency of the mixed-transportation pump was small. $\Delta Pd/Pd \leq 0.01$ and $\Delta\eta/\eta \leq 0.001$, where $Pd1$ is the boost pressure under mesh I, $\Delta Pd$ is the difference between the boost pressure calculated using each group of mesh I and the boost pressure corresponding to mesh I. $\eta_c$ is the efficiency value calculated via simulation under mesh I, $\Delta\eta$ is the difference between the efficiency corresponding to each group of mesh I and the efficiency corresponding to mesh II. Finally, in the following simulation, mesh IV with a total grid number of 7,734,156 was selected as the final grid for numerical calculations.

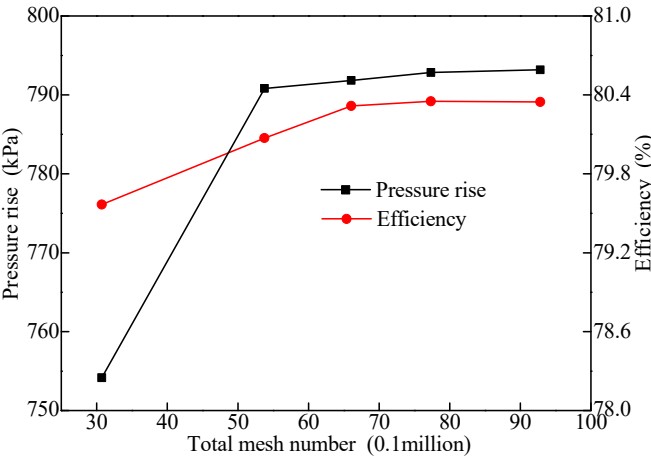

**Figure 4.** Mesh validation.

### 3.3. Experimental Verification

The experimental system platform of the multistage mixed-transport pumps is shown in Figure 5. This platform was used to check the reliability of the numerical calculation results through the external characteristic test of the mixed-transport pump under different IGVFs. The experimental equipment and instruments met the level 2 accuracy requirements of GB/T 3216-2016. The liquid flow rate was controlled at a certain constant value, and the gas flow rate into the main pipeline was controlled using a regulating valve. Thus, the gas–liquid mixed medium with different gas contents was achieved. The external performance of the multistage mixed-transport pump under diverse gas contents was measured via the multiphase flow test. The main components and measuring elements of the experimental system are presented in the figure as well. The gas–liquid mixer 9 ensured that the mixture medium entering the inlet of the mixing pump was sufficiently mixed to form a bubble inflow condition, thereby ensuring the accuracy of the experiment.

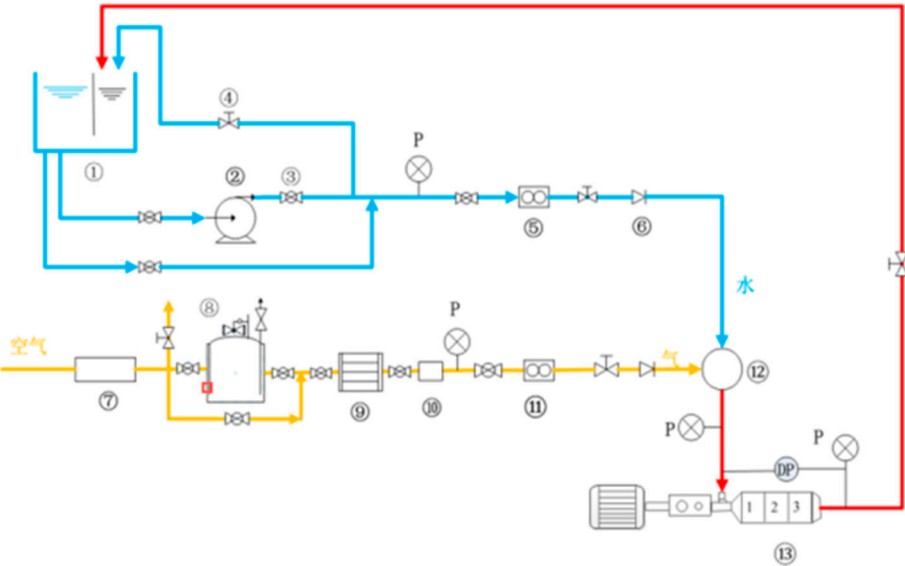

**Figure 5.** Experimental system schematic diagram of the multiphase pump. Labels: P—Pressure sensor; DP—Differential pressure sensor, 1—Tank; 2—Pressure pump; 3,4—Valves; 5—Mass flowmeter; 6—Check valve; 7—Air compressor; 8—Gasholder; 9—Air dryer; 10—Filter; 11—Mass flowmeter; 12—Commingler; 13—Multiphase pump test section.

Figure 6 shows the comparison between the numerical simulation results and the experimental results under different flow conditions of pure water at 3500 rpm. The numerical calculation results

were very consistent with the data from the experiment. Thus, the numerical simulation method and calculation accuracy had a high reliability and accuracy. Under the pure liquid condition, the head decreased gradually with an increased flow rate, and a hump appeared at 0.32$Qd$. When the volume flow rate was larger than 0.37$Qd$, the efficiency growth rate of this pump increased slowly. As such, with an increased flow rate, the effects of the flow rate on efficiency were limited. The deviation between the experimental and numerical results may have been caused by not considering the roughness of the wall surface and ignoring the leakage loss at the impeller inlet, the loss along the pipeline, and the local loss. Furthermore, the differential pressure sensor used for measurement was a certain distance from the inlet position and outlet side of the mixed-transport pump, while the flow velocity in the pipeline increased with an increasing flow rate, the resistance loss along the pipeline and the local loss increased. Therefore, the larger the flow rate, the greater the error of the experimental and numerical results.

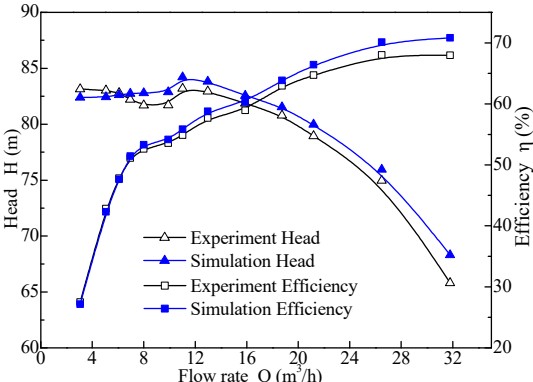

**Figure 6.** Head and efficiency performance in the simulation and experiment.

Table 3 shows the head values comparison between the numerical simulation and the experiment of the multiphase mixed-transport pump at the rated flow rate under different air content conditions. With increasing IGVFs, the head value of the mixed-transport pump decreased continuously. Moreover, the error value between the numerical simulation and experiment increased, thereby indicating that the gas content had a considerable influence on the characteristics of the mixed-transport pump. When the inlet gas content was less than 20%, the numerical results were good agreed with the experimental results, and the relative deviations were within the allowable region. When the IGVF was greater than 20%, the numerical simulation result deviated remarkably from the experimental value. Thus, the numerical calculation method was suitable for the numerical simulation when the IGVF was less than 20%. The numerical calculation method under a high gas content should be further studied.

**Table 3.** Head data of the simulation (CFD) and experiment (EXP) under different inlet-gas-volume fractions (IGVFs).

| Conditions | Head_CFD (m) | Head_EXP (m) | Variation (%) |
| --- | --- | --- | --- |
| IGVF = 0 | 79.63 | 78.82 | 1.03% |
| IGVF = 5% | 74.75 | 73.55 | 1.63% |
| IGVF = 10% | 65.28 | 63.42 | 2.93% |
| IGVF = 15% | 53.79 | 51.71 | 4.02% |
| IGVF = 20% | 43.25 | 41.26 | 4.82% |
| IGVF = 25% | 35.05 | 32.63 | 7.41% |

## 4. Numerical Results and Analysis

### 4.1. Pressure Distribution

Figure 7 shows the pressure distribution in the second stage impeller and diffuser of the multistage mixed-transfer pump at 10%, 50%, and 90% blade height under a 20% IGVF. The pressure distribution

at the front and back of the blades at different heights was uniform but the pressure of the impeller blade varied with different blade heights. This result shows that the supercharging capacity of the impeller was different from the front cover to the back cover but the change trend of the pressure gradient in each passage of the impeller was similar. Thus, the design of the impeller was reasonable. The pressure at the front cover of the impeller blades was greater than that at the back cover, indicating that the pressure-boosting capacity at the front cover was stronger than that at the back cover. The pressure variation at the inlet of the first stage impeller and diffuser changed sharply, directly leading to a large pressure pulsation at the blade head of the impeller and diffuser. This result was consistent with the results of the pressure fluctuation frequency domain diagram.

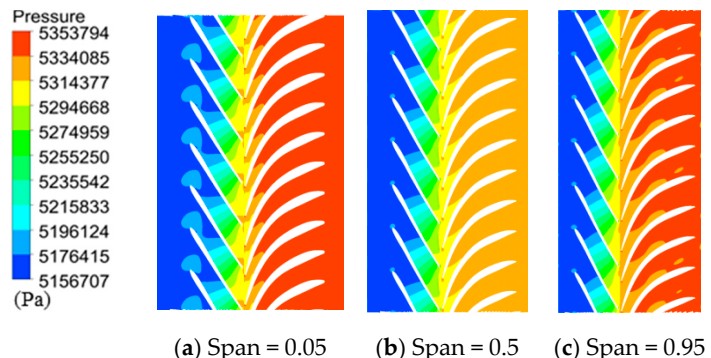

(**a**) Span = 0.05        (**b**) Span = 0.5        (**c**) Span = 0.95

**Figure 7.** Pressure distribution at diverse spans under IGVF = 20%.

### 4.2. Pressure Pulsation Analysis

The designed speed of the impeller in the mixed-transport pump was 3500 rpm, and its frequency was F0 = 58.33 Hz. The frequency domain diagram of the pressure fluctuation of the impeller passage monitoring locations IMS1–IMS5 and DPS1–DPS5 in the diffuser passage under different IGVF working conditions is shown in Figure 8. When the IGVF was different, the change in the pressure pulsation amplitude at respective monitoring point was different. Given the work done by the impeller, the pressure increased from the impeller inlet to the diffuser outlet gradually. The main frequency amplitude of the pressure pulsation in the diffuser was greater than that in the impeller, and the pressure pulsation amplitude reached the maximum value at the impeller blade head and the middle position of the diffuser passage, which was related to the gas phase aggregation. This result is consistent with the results of gas phase distribution in the impeller. Comparison of the pressure pulsation frequency domain graphs under different gas content conditions was undertaken, and the results indicate that the main frequency amplitude increased with an increased gas content. Furthermore, the amplitudes of IMS1, IMS5, DPS1, and DPS5 were higher than other points due to the uneven distribution of the phase state and the flow impact caused by the interference between the rotating part (impeller outlet) and the stationary part (diffuser inlet). In most cases, the dominant frequency of the impeller was 583.3 Hz, and was affected by the diffuser vanes number, whose value is the product of the rotation frequency and number of guide vanes. The same situation happened in the diffuser, where the dominant frequency value was 408.3 Hz and was seven times of the rotation frequency, which is a multiple equal to the number of impeller blades. Therefore, we showed that the dominant frequency of the pressure pulsation in the diffuser and impeller was affected by the number of blades in each and was an integer multiple of the rotation frequency.

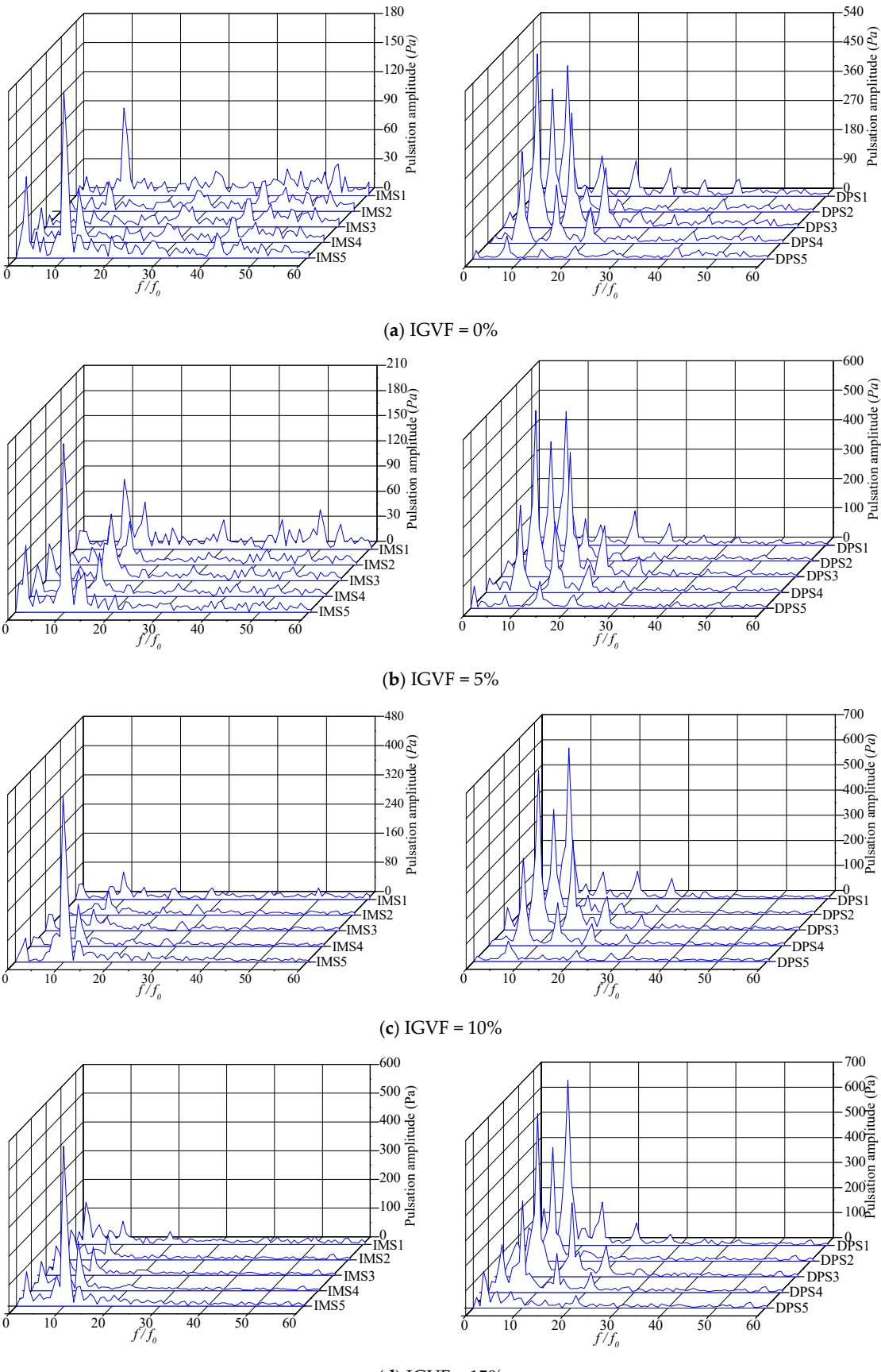

(**a**) IGVF = 0%

(**b**) IGVF = 5%

(**c**) IGVF = 10%

(**d**) IGVF = 15%

**Figure 8.** *Cont.*

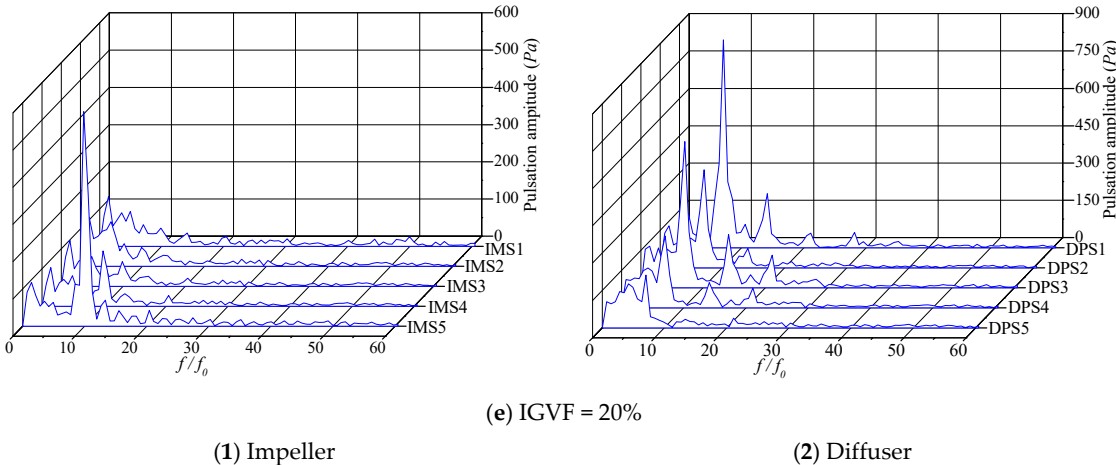

(**e**) IGVF = 20%

(**1**) Impeller                    (**2**) Diffuser

**Figure 8.** Pressure fluctuation frequency domain diagrams for IMS1–IMS5 and DPS1–DPS5.

*4.3. Gas Phase Distribution*

Figure 9 shows the air volume fraction distribution at different spans of the blade heights in the second stage impeller and diffuser of the mixed-transport pump with a 20% inlet gas content. The gas phase in the impeller flow channel was primarily gathered in the suction surface of the front cover plate at the impeller outlet. The air volume fraction near the front cover plate was greater than the back cover plate. Given the work done by the impeller, the bubbles were removed toward the exit direction and were influenced by the differential pressure between the front and rear of the blade. This phenomenon further promoted the gas movement toward the back surface of the blade. Given that the density of the gas and liquid phase was different, the water was higher than the air; therefore, it was subjected to a greater centrifugal force. This force pushed the water to the back cover plate. At the impeller blade outlet, the GVF of the local area was higher due to liquid phase flow separation, which also caused the increase in pressure fluctuation amplitude of the IMS1 monitoring point. In the diffuser channel, the gas phase primarily gathered at the front cover and gradually diffused to the main flow area along the flow orientation. The gas at the outlet of the impeller passage continued to flow into the diffuser passage due to an inertia effect. The centrifugal force effect disappeared in the diffuser passage, and the density of air was less than water. As such, the gas phase diffused to the main flow area and moved downstream in the guide vane passage.

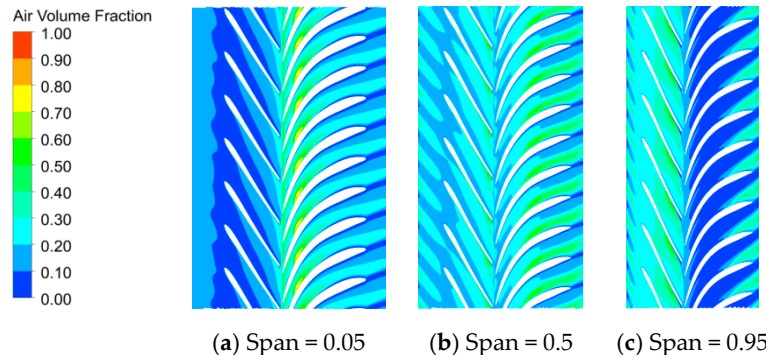

(**a**) Span = 0.05        (**b**) Span = 0.5        (**c**) Span = 0.95

**Figure 9.** GVF distribution at diverse spans in the second stage impeller and diffuser under IGVF = 20%.

The distribution of the gas content on the surface of the second impeller under the conditions of GVF = 5%, 10%, 15%, and 20% is shown in Figure 10. With an increased IGVF, the air aggregation area on the suction surface of blades increased continuously within the four different IGVF situations. The gas phase primarily gathered in the suction surface of the blade head. The results of the GVF distribution show that the gas distribution at the blade head was relatively high, thereby affecting the

amplitude of the pressure pulsation at the blade head, and the values of the amplitude at blade head were the largest. This phenomenon was attributed to the mixed medium that entered the impeller and collided with the blade head. Another reason was the influence of the impeller rotation. The gas phase was primarily gathered close to the front cover of the impeller given that the water within a higher density is subjected to more centrifugal force than the gas phase; thus, the pressure on the rear cover plate was greater than that on the front cover. A pressure difference between the impeller shroud and hub was observed; thus, the gas content at the hub was greater than that at the shroud. With an enhanced gas content, the air in the impeller passage moved toward the outlet along the blade surface. Under a high GVF condition, the air accumulated in the first half of the impeller passage, and a large amount of air blocked a portion of the flow channel. The air was affected by the pressure gradient, which caused the air bubbles to flow in the opposite direction at the outlet position of impeller blade suction surface.

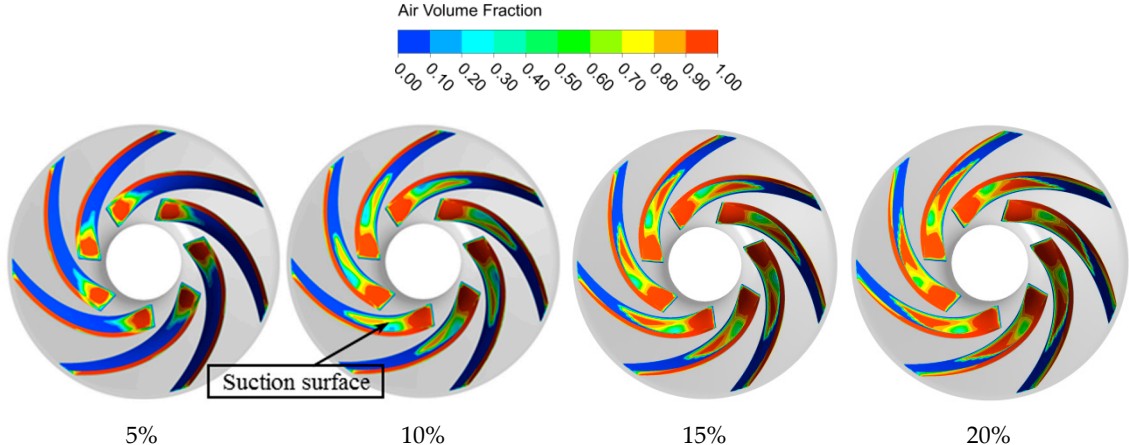

**Figure 10.** Distribution of GVF on the blade of the second impeller for different gas contents.

### 4.4. Liquid Relative Velocity Distribution

The distribution diagram of the inter-blade blade relative liquid velocity at 10%, 50%, and 90% heights of the blades in the second stage mixed-transport pump under the IGVF = 20% condition is shown in Figure 11. The liquid velocity distribution at different heights of the blades was different and related to the pressure distribution and IGVF. Given the energy input of the pump shaft, the liquid velocity in the impeller flow passage increased stage-by-stage along the radius orientation, and the liquid velocity distribution was relatively uniform. A small low-velocity region existed on the suction side near the impeller blade exit edge, which was related to the volume distribution of the gas phase. Because of the influence of the centrifugal force, the liquid phase flow separation, and the gas aggregation at the suction surface near the blade outlet, the low-velocity zone of liquid phase was formed. The liquid low-velocity region increased from the front cover to the rear cover, thereby indicating that the gas phase was primarily gathered in the shroud of the impeller. However, the liquid velocity decreased gradually after entering the diffuser due to the action of the diffuser. The kinetic energy was gradually converted into energy pressure. Moreover, after the gas–liquid mixed medium entered the diffuser passage, the velocity of the mixed medium affected by the impeller still existed. Therefore, the velocity of the guide vane pressure surface was greater than the reverse surface. The liquid-phase velocity increased gradually from the front cover to the rear cover within the diffuser. Given the effects of the impeller in the next stage, a low-velocity region of the liquid phase appeared close to the outlet of the fixed vane, which resulted from static and dynamic interference.

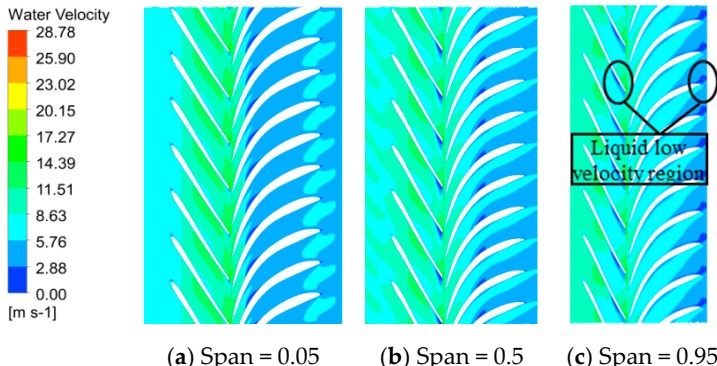

(**a**) Span = 0.05    (**b**) Span = 0.5    (**c**) Span = 0.95

**Figure 11.** Water velocity distribution at diverse spans in the second impeller and diffuser under IGVF = 20%.

The streamline distribution in the second-stage impeller and diffuser of the mixed-transportation pump under the different inlet gas content is shown in Figure 12. On the whole, the internal flow lines of each impeller were relatively smooth and smooth under diverse gas content conditions. Thus, the impeller exhibited good hydraulic performance under pure water and air and water two-phase conditions. The vortices in the middle part of the diffuser flow channel near the guide vane back surface were obvious. The fixed guide vane diffuser of the multi-stage mixed-transport pump was directly connected to the upper and lower parts, the flow passage was long, and the fixed guide vane wrap angle was too large, bringing about a flow separation and vortex generation. Compared with the streamline distribution in the diffuser with different GVFs, the liquid velocity in the flow passage increased with increasing IGVFs. The vortex position tended to move toward the guide vane inlet of the diffuser with increasing GVFs.

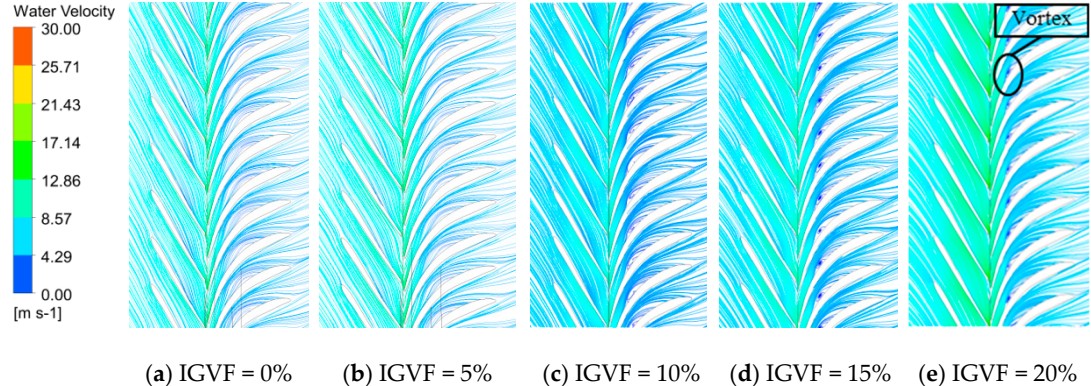

(**a**) IGVF = 0%    (**b**) IGVF = 5%    (**c**) IGVF = 10%    (**d**) IGVF = 15%    (**e**) IGVF = 20%

**Figure 12.** Streamline distribution in the second-stage compression unit with different IGVFs.

## 5. Conclusions

The experimental results validated the accuracy of the numerical simulation process of the multiphase transport pump. The pressure fluctuation and gas phase distribution in the compression unit and the impeller of this pump at a rated flow with diverse IGVFs were studied. The conclusions were as follows:

(1)   For the same experimental and numerical simulation conditions, the pressurization capacity of the multiphase mixed-transport pump decreased gradually with increasing flow rate. The efficiency improvement of this pump was limited with increasing flow rates. Meanwhile, the relative discharge of the IGVF and the decrease of the pump head had a quantitative relationship under a rated-flow condition. The pressurization capacity of the mixed-transport pump decreased with increasing the IGVFs. At the equal flow rate condition, the pressurization capacity of the mixed pump decreased by more than 30% when the IGVF was greater than 13.5%.

(2)    The amplitude of the pressure pulsation close to the impeller blade head suction surface increased gradually along the flow orientation. With increasing IGVFs, the pressure fluctuation intensity at the impeller and diffuser inlet increased significantly. This phenomenon was closely correlated with the gas accumulation and the low relative velocity area.

(3)    The main frequency amplitude increased continuously with increasing gas content. In most cases, the dominant frequency of the impeller was 583.3 Hz, which was affected by the diffuser vanes number, whose value is the product of the rotation frequency and the number of guide vanes. The same situation happened in the diffuser, where the dominant frequency value was 408.3 Hz and was seven times that of the rotation frequency. It was shown that the pressure pulsation dominant frequency was an integer multiple of the rotation frequency.

(4)    The gas in the impeller passage primarily gathered on the suction surface of the blade head close to the front cover plate. Due to the work of the centrifugal forces, the water was pushed to the back cover plate, increasing the GVF near the front cover plate. On the blade suction surface near the front cover plate, a low-velocity area caused by the flow separation was generated. Thus, the pressure pulsation of the impeller was further affected. Vortices in the diffuser were obvious, and the vortex position tended to move toward the inlet of the diffuser with increasing gas content.

**Author Contributions:** Investigation, numerical simulation, data analysis, and paper writing were carried out by C.L.; X.L. and J.F. conceived and supervised the study and edited the manuscript; and G.Z. and Y.X. wrote the manuscript. All authors reviewed the manuscript. All authors have read and agree to the published version of the manuscript.

**Funding:** This study was supported by the National Key R&D Program of China (No. 2018YFB1501900), the National Natural Science Foundation of China (NSFC, No. 51527808, No. 51679195 No. 51769012, and No. 51909212), the Natural Science Basic Research Plan in Shaanxi Province of China (No. 2019JQ-044), and the Scientific Research Plan Projects of Shaanxi Education Department (No. 19JK0570).

**Conflicts of Interest:** The authors declare no conflict of interest.

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
