# Peer review of "Effects of Gas-Volume Fractions on the External Characteristics and Pressure Fluctuation of a Multistage Mixed-Transport Pump"

_applsci, doi:10.3390/app10020582_

Round 1

Reviewer 1 Report

This report describes the simulation and experimentation of pump efficiency to transport oil/gas mixture.

This manuscript is fundamentally well written and potentially useful in this field.  However, the authors must address the following issues before the publication.

1.

In Fig. 6, why the simulation is lower than the experiment in low flow rate condition?

2.

Fig. 5 picture is messy.  A more easily understandable schematic illustration should be added.

3.

From the reviewer’s expertise, the scale effect of this analysis is interesting.  For example, micrometer scale pump with peristaltic principle is reported (Micromachines 7(5), 83 (2016)) for liquid-gas mixture.  Is the analysis also effective for this micrometer-scale if the impeller or diffuser can be integrated?  Authors should also discuss about this point if possible.

Author Response

Thanks for the excellent comments of the reviewer. The corresponding replies and modifications are detailed as the following: The revised manuscript shows the new/changed text using track changes.

Review report (Round 1)

In Fig. 6, why the simulation is lower than the experiment in low flow rate condition?

Reply: Thanks for your suggestion! Under low flow rate conditions, due to the decrease of liquid flow, the gas entering the mixed transport pump increases, resulting in the aggravation of the flow separation phenomenon. However, the numerical simulation excessively predicts flow separation; which leads to the simulation results are lower than the experimental data under the low flow rate conditions.

Fig. 5 picture is messy. A more easily understandable schematic illustration should be added.

Reply: Thanks for your suggestion! The authors has modified figure 5 in the article, given a schematic diagram of the experimental system, and highlighted them in yellow in the article page 6 figure 5 and lines 187-189.

From the reviewer’s expertise, the scale effect of this analysis is interesting. For example, micrometer scale pump with peristaltic principle is reported (Micromachines 7(5), 83 (2016)) for liquid-gas mixture. Is the analysis also effective for this micrometer-scale if the impeller or diffuser can be integrated? Authors should also discuss about this point if possible.

Reply: Thanks for your suggestion! The author read Yaxiaer Yalikun and Yo Tanaka's article "Large - Scale Integration of All - Glass Valves on a Microfluidic Device (Micromachines 7(5), 83 (2016)" carefully. The large-scale integrated flow device described in that paper is different from the author’s research field. Yaxiaer’s research on the microfluidic device as a peristaltic pump under liquid-gas mixture flow is to measure fluid velocity through bubble flow, and the flow rate is very small. However, this paper studied the effect of gas volume fraction on the mixed transport pump performance. Therefore, the scale effect of this analysis may be not effective for micrometer-scale pump.

Thanks again for the reviewer’s Suggestions, which provided some help and direction for the author's follow-up research.

Reviewer 2 Report

In this article the performance and pressure pulsation of a of a three-stage mixed transport pump with different inlet-gas-volume fractions (IGVFs) were studied, both numerically and experimentally. In particular, it was shown that IGVF greatly influences the dominant frequency of pressure fluctuation, with good agreement between simulation and experiments. These results are useful, as they may be used to optimize the design of multiphase pumps.

The article is interesting and well written, and I recommend it for publication.

My only observation is the following. In Table 1 we see that Q, measured in m^3/h, denotes a volume flow rate, in agreement with a dimensional analysis of Eq. (5). However, in lines 162 and 194, it is written that Q is a mass flow rate. This should be corrected.

Author Response

Thanks for the excellent comments of the reviewer. The corresponding replies and modifications are detailed as the following: The revised manuscript shows the new/changed text using track changes.

Review2 report (Round 1)

Comments and Suggestions for Authors

In this article the performance and pressure pulsation of a of a three-stage mixed transport pump with different inlet-gas-volume fractions (IGVFs) were studied, both numerically and experimentally. In particular, it was shown that IGVF greatly influences the dominant frequency of pressure fluctuation, with good agreement between simulation and experiments. These results are useful, as they may be used to optimize the design of multiphase pumps.

The article is interesting and well written, and I recommend it for publication.

My only observation is the following. In Table 1 we see that Q, measured in m^3/h, denotes a volume flow rate, in agreement with a dimensional analysis of Eq. (5). However, in lines 162 and 194, it is written that Q is a mass flow rate. This should be corrected.

Reply: Thank you for your suggestion! About the representation of flow rate Q, the authors have modified in lines 162 and194 in the article and highlighted them in yellow.
